# A Dual-branch Multi-Band Neural Vocoder with Harmonic Discriminator for High-Fidelity Speech Synthesis

## Abstract

Recent developments in vocoders are primarily dominated by GAN-based networks targeting to high-quality waveform generation from mel-spectrogram representations. However, these methods are typically computationally expensive and operate in the time-domain which neglect the time-frequency structures. In this paper, we propose the DMNet, a Dual-branch Multi-band Network to address these limitations. First, a reconstruction network of complex-valued spectrogram called CondNet is used as a condition and thus integrated into the GAN-based branch. Second, we use multi-band processing in the dual-branch: the Cond-Net produces Fourier spectral coefficients in one sub-band signal and GAN-based branch generates sub-band representations which are subsequently transformed to full-band speech. Finally, to further improve fidelity, we propose a novel harmonic discriminator which utilizes learnable harmonic filters at multiple scales for a better modeling ability in harmonic structures. In our experiments, DMNet validates the effectiveness and achieves superior performance for high quality waveform generation, both on subjective and objective metrics.

## 1 Introduction

In the real world, speech is an extremely important modality for various practical applications. Neural network based vocoders aiming to generate high quality waveform from an intermediate representation play a crucial role in speech or audio synthesis. In particular, mel-spectrograms which have approximate human auditory perceptions and compact dimensionality are widely used as the intermediate representations, especially in text-to-speech (TTS) (Ren et al., 2019; Ma et al., 2019; Li et al., 2023e; Jiang et al., 2024; Du et al., 2024), singing voice synthesis (SVS) (Liu et al., 2022a; He et al., 2023; Lei et al., 2023; Zhang et al., 2024) and voice conversion (VC) (Qian et al., 2019; Li et al., 2023c;d; Ning et al., 2023; 2024) technologies. A two-stage strategy is always used in these methods: the intermediate mel-spectrogram representation is first predicted from source feature and next stage converts it into a raw waveform. The traditional signal processing approaches mainly focus to map intermediate feature to the original speech, which introduces nonnegligible artifacts. In recent years, with the success of deep learning, mel-spectrogram based neural vocoders have been rapidly improved in the aspect of quality and naturalness of speech.

Generative adversarial network based neural vocoders are one family of methods that are the most effective and efficient so far. These methods usually employ convolutional neural network (CNN) architectures with temporal transposed convolution layers to directly sequential upsample the mel-spectrogram representation to the raw waveform. However, such a black box operation introduces some problems. First, modeling the raw waveform with the high temporal resolution (e.g., 16,000 samples per second for 16k Hz) is a notably challenging problem. Second, the redundant upsampling convolution processes in the temporal domain are impressionable to aliasing artifacts. Although models like BigVGAN (Lee et al., 2023) alleviate the impacts by anti-aliased multi-periodicity composition module, they are parameter-heavy and computationally expensive. The last but the most important, the mel-spectrogram based vocoders have three inverse problems to be solved as Kaneko et al. (2022) states, i.e., recovery of the Fourier spectral coefficients (magnitude and phase spectrums) and frequency-to-time conversion. GAN-based vocoders directly generate a raw wave-

Table 1: The illustrations of primary properties for different vocoders considered in human speech.

| Model Type | Generator | | Discriminator | | Speed |
| --- | --- | --- | --- | --- | --- |
| | time-frequency | sub-band harmonic | fixed resolution | dynamic resolution | |
| GAN | × | × | ✓ | × | slow |
| ISTFT | ✓ | × | ✓ | × | fast |
| DMNet | ✓ | ✓ | ✓ | ✓ | fast |

form in the temporal domain, which cannot capture the well-existed inductive time-frequency bias in the mel-spectrogram representation.

The recent inverse short-time Fourier transform (iSTFT) based vocoders gradually become an appealing alternative of GAN-based methods due to their closeness to human auditory perception and well-established fast computation. However, these systems usually predict the high-dimensional full-band Fourier spectral coefficients, i.e., magnitude and phase spectrums, and next apply iSTFT to generate the high-quality waveform, which is a challenging area of waveform reconstruction. Generally, a superior waveform generation network architecture always needs redundant estimations of full-band magnitude and phase spectrums. In addition, phase spectrum in the time-frequency domain has the complicated nature and periodic-related structure resulting in the principal values wrapping across all phase angles (Siuzdak, 2024). Furthermore, harmonic property of speech plays a critical role and thus must be considered in the full-band signal reconstruction process. While some researchers have explored to predict the harmonic or pitch of speech (Wang et al., 2019; Engel et al., 2020; Łańcucki, 2021; Li et al., 2023a;b; Subramani et al., 2024), challenges still remain in the aspect of the harmonic modeling in terms of full-band signals.

To address the aforementioned limitations, we propose a **D**ual-branch **M**ulti-band **Net**work architecture with a harmonic discriminator for the neural vocoder, named **DMNet**. In this paper, a multi-band GAN-based branch integrating with the spectrum reconstruction based CondNet in the time-frequency domain is designed to learn the sub-band related information, and a harmonic discriminator is used to further enhance the harmonic structures of speech. The main illustrations among these methods are summarized in Table 1. Specifically in our work, we first utilize a modified TFGAN network (Tian et al., 2020) as the GAN-based branch to generate multiple sub-band signals. In addition, a spectrum reconstruction network is used to predict Fourier spectral coefficients and further converts these coefficients to the waveform. Furthermore, the reconstructed waveform and time-frequency representation are used as prior conditions to integrate into the corresponding upsampling layers in GAN-based branch by the convolutional blocks and hybrid downsample modules. The GAN-based vocoder will benefit from these operations by facilitating the fusion of inductive bias in the time-frequency domain. Second, we employ multi-band processing in the dual-branch network, which lifts the computational cost. Other than producing multiple sub-band signals by GAN network, we predict the magnitude and phase spectrums in one low-frequency sub-band rather than the full-band in the CondNet, which avoids the complex multi-periodicity (harmonic) components modeling in the full-band and further mitigates the aliased artifact and improves the inference speed. Finally, we design a multi-scale harmonic discriminator with the learnable triangular band-pass filters. Instead of a fixed time-frequency resolution of the discriminator like Jang et al. (2021) and Défossez et al. (2022), the harmonic discriminator owns dynamic frequency resolution, achieving the significant advantages for better harmonic tracking.

Specifically, the main contributions in our paper are as follows:

- We propose the DMNet, a dual-branch multi-band based vocoder, trained to generate the raw waveform with a fusion of time and time-frequency domain information. A conditional reconstruction network, termed CondNet, is used to predict Fourier spectral coefficients. Unlike previous iSTFT-based networks that rely on full-band Fourier spectral coefficients, this CondNet predict these only in one sub-band, preserving the primary low-frequency bands and contributing to better performance and faster inference speed.

- To further improve the speech fidelity, a novel multiple scale based harmonic discriminator with dynamic resolution in frequency domain is proposed. For the discriminator network architecture, we consider the harmonic as channel axis and thus utilize the depthwise sepa-

rable convolution and pointwise convolution to learn the intra-harmonic and inter-harmonic property of speech, respectively.

- Our extensive experiment results demonstrate that DMNet matches competitive speech quality in terms of subjective and objective metrics. In addition, we also validate the effectiveness of the proposed CondNet. Furthermore, our DMNet achieves the superior inference speed compared with other baseline methods especially conventional GAN-based vocoders due to the multi-band signal processing.

The rest of our proposed paper is organized as follows: In Section 2, some related vocoder research methods are introduced. Next in Section 3, we introduce the proposed vocoder method, including the dual-branch multi-band network architecture and the harmonic discriminator. Experimental results are reported in Section 4. Finally, Section 5 is the conclusion of our paper.

## 2 RELATED WORK

**GAN-based vocoder.** Compared with conventional vocoders, GAN-based vocoders are gaining growing attention due to their efficient ability for waveform generation. HiFiGAN (Kong et al., 2020) is the typical GAN-based method, which utilizes the multi-receptive field fusion (MRF) module for better performance. In addition, progress is further achieved by introducing the multi-period discriminator (MPD) (Kong et al., 2020) and the multi-scale discriminator (MSD) (Kumar et al., 2019). MelGAN (Kumar et al., 2019) utilizes a non-autoregressive fully convolutional feed-forward architecture for waveform generation without additional perceptual loss functions. TFGAN (Tian et al., 2020) is an extension of MelGAN in terms of generator, and employs the time discriminator and frequency discriminator to offer more consistency guarantees, respectively. Moreover, a set of time-domain loss is used for better waveform quality. Avocodo (Bak et al., 2023) jointly optimize a sub-band discriminator and a collaborative multi-band discriminator to alleviate unintended artifacts. In particular, BigVGAN (Lee et al., 2023) achieves the state-of-the-art synthesis quality of speech with the periodic activations and anti-aliased multi-periodicity composition (AMP) module in the generator. Although GAN-based vocoders achieve the high fidelity, the inductive bias in time-frequency mel-spectrogram is not well utilized, which degrades the synthesized waveform quality to a certain extent.

**ISTFT-based vocoder.** Another explored neural vocoder is iSTFT-based network architecture. These systems usually reconstruct waveform by parameterizing the model to predict full-band Fourier spectral coefficients, i.e., phase and magnitude components. The iSTFTNET (Kaneko et al., 2022) and iSTFTNET2 (Kaneko et al., 2023) are a series of researches that make some modifications of HiFiGAN. Some upsample blocks with transposed convolutions are replaced with inverse STFT in order to return Fourier spectral coefficients. In addition, some iSTFT-based vocoders are explored without upsample blocks. HiNet (Ai & Ling, 2020) utilizes an amplitude spectrum predictor (ASP) to predict amplitude and a NSF-based (Wang et al., 2019) phase spectrum predictor (PSP) for phase prediction. ASP and PSP modules are hierarchical orders for prediction. In other words, the amplitude is first generated and next is the phase spectrum generation. APNet (Ai & Ling, 2023) and APNet2 (Du et al., 2023) design the ASP and PSP modules as parallel structures and multi-level loss functions are used for better optimization. Furthermore, Vocos (Siuzdak, 2024) treats the magnitude and phase predictions as a whole block, which employs ConvNeXt (Liu et al., 2022b) blocks with Gaussian Error Linear Unit (GELU) (Hendrycks & Gimpel, 2016) activations to predict magnitude and phase components simultaneously. All the aforementioned methods use the inverse STFT to reconstruct the full-band waveform. Unfortunately, the multi-periodicity components are always existent in the full-band signals, bringing challenges in the harmonic modeling and aliasing artifacts elimination.

## 3 METHOD

In this section, our proposed DMNet architecture will be introduced. To begin with, we introduce the overview of the proposed model. Next, we provide detailed introductions of the conditional spectrum reconstruction network, i.e., CondNet, the multi-band processing as well as the proposed harmonic discriminator. Finally, we introduce the training objectives.

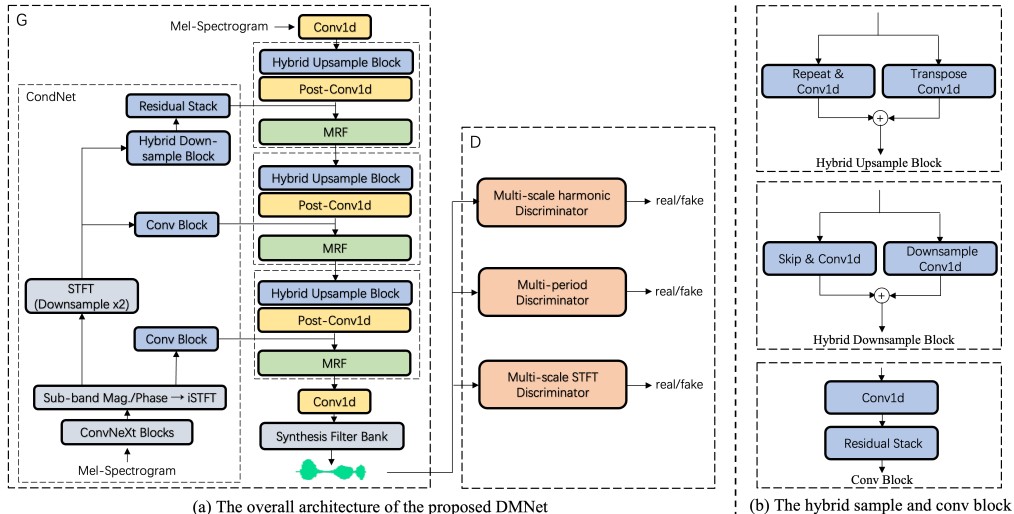

(a) The overall architecture of the proposed DMNet

(b) The hybrid sample and conv block

Figure 1: The overall architecture of the proposed DMNet and the hybrid sample block. In figure (a), the generator and discriminator are shown, respectively. In the generator, the CondNet is shown in the left and ConvNeXt blocks are used to predict the magnitude and phase in one sub-band. Then, convolutional and hybrid downsample blocks are utilized for condition integrations. MRF denotes the Multi-Receptive Field Fusion module. In figure (b), the detailed architectures of the hybrid upsample and downsample blocks, as well as convolutional block are shown, respectively.

## 3.1 OVERVIEW

As illustrated in Figure 1, the proposed model is composed of generator and discriminator. In terms of the generator, the GAN-based backbone network is a modified TFGAN. Specifically, the sinusoidal activation function is removed and the residual stacks are replaced by MRF (Kong et al., 2020). A post-convolution layer is used as the lightweight bottleneck between the upsample block and MRF for information aggregation. Next, in order to learn the time-frequency structures, a spectrum reconstruction network is used to predict magnitude and phase components in one sub-band signal. The sub-band waveform is generated by an inverse STFT operator. Then, the reconstructed conditional waveform and time-frequency representation are used to integrate into the backbone network to generate the final sub-band signals. The final waveform is generated by a pseudo-QMF (PQMF) bank (Nguyen, 1994) without distortion. In our work, we only use the synthesis filter bank in PQMF to generate waveforms. As for the discriminator, we utilize the proposed harmonic discriminator, the multi-period discriminator (MPD) (Kong et al., 2020) as well as the multi-scale STFT-based (MS-STFT) discriminator (Défossez et al., 2022) to train our model.

## 3.2 CONDNET

As the black box GAN-based generator network is lack of the guidance in the time-frequency domain and iSTFT-based network can offer the inductive time-frequency bias, we propose a conditional network, i.e., CondNet as shown in Figure 1 to compensate the loss. First, we utilize the ConvNeXt (Liu et al., 2022b) blocks to predict magnitude and phase components in one sub-band simultaneously from the mel-spectrogram representation. We employ the exponential function to represent the magnitude and apply the cosine and sine to denote the real and imaginary parts, respectively. These complex-valued Fourier spectral coefficients in one sub-band learning the time-frequency structures are used to convert into the waveform with the inverse STFT operator. Next, we couple the predicted time-domain signal of spectrum reconstruction network with the GAN-based generator branch more effectively. Specifically, for the first layer, we directly feed the sub-band waveform to the convolutional block. Next, we initially perform the STFT operator for $2\times$ downsampling, and then adopt the convolutional block and hybrid downsample module (Shi et al., 2024) to hierarchically integrate into the second and third layers (Figure 1). Finally, we directly add these time and time-frequency domain representations to each upsample block output respectively as the prior condition knowledge to guide the final multi-band signals learning.

## 3.3 Multi-band Processing

In this section, we will introduce the multi-band processing in the dual-branch network. For the GAN-based generator, we directly predict the multi-band representations, i.e., the final output channel $C$ stands for the number of sub-band signals. The total upsampled multiple reduces by $C$ times compared with the original full-band generator, further resulting in the reduction in the time axis (e.g., 24000 samples in one second, output channel is 12, thus the output length is 2000). In terms of the CondNet, Fourier spectral coefficients in one low-frequency sub-band are predicted and the hop size of the inverse STFT also can reduce by $C$ times (e.g., the hop size of full-band is 240, the number of sub-band is 12, thus the hop size is only 20 in sub-band). In one low-frequency sub-band, the CondNet can neglect the multi-periodicity and harmonic components of the full-band signal, which is beneficial for model training. The final waveform is generated by the effective PQMF bank. Furthermore, the whole dual-branch generator network is trained in the form of sub-band, significantly improving the inference speed.

## 3.4 Harmonic Discriminator

The GAN-based methods usually apply the discriminators for adversarial training. However, these discriminators are typically designed in a fixed time-frequency resolution manner. In the real world, many signals require to focus different frequency bands such as music or singing voices. Therefore, we design a novel harmonic discriminator with dynamic frequency resolution by applying the learnable band-pass filters, which is a first-of-its-kind approach.

Motivated by Bittner et al. (2017); Ravanelli & Bengio (2018); Won et al. (2020), a triangular band-pass filter $\Delta$ can be written as follows:

$$\Delta\left(f; f_c; f_w\right) = \left[1 - \frac{2\left|f - f_c\right|}{f_w}\right]_+,  \tag{1}$$

where $f$ is the frequency bin, $f_c$ and $f_w$ are the center frequency and bandwidth, respectively. $[.]_+$ stands for the rectified linear function. In addition, the bandwidth can be empirically formulated as a function of the center frequency $f_c$: $f_w \cong 0.1079 f_c + 24.7$. In our paper, we use the learnable parameters $\alpha$, $\beta$ and $\sigma$ to compute the bandwidth: $f_w = (\alpha f_c + \beta)/\sigma$. Therefore, as the center frequency $f_c$ goes higher, the bandwidth $f_w$ goes wider, which means a higher time resolution in the high-frequency band and further improves the tracking ability of fast-changing harmonics.

Furthermore, the center frequency $f_c$ at the $n$-th frequency bin of the first harmonic filter can be defined as follows:

$$f_c^n = f_{min} \cdot 2^{\frac{n}{B}},  \tag{2}$$

where the minimum frequency $f_{min} = 32.7$ Hz (C1), $B$ is the number of bins per octave ($B = 24$ in our paper) and $n$ denotes the filter index. The maximum frequency in the first harmonic is limited as $f_{max} = f_s/2K$ to satisfy the Nyquist criterion, where $f_s$ is the sampling rate of speech and $K$ denotes the number of harmonics.

Following the above definition, a harmonic filter bank can be defined as follows:

$$\Delta_k\left(f; f_c; \alpha, \beta, \sigma\right) = \left[1 - \frac{2\left|f - k \cdot f_c\right|}{(k \cdot \alpha f_c + \beta)/\sigma}\right]_+,  \tag{3}$$

where $k$ stands for the index of $k$-th harmonic with the center frequency $f_c$. Therefore, the harmonic filter bank with $K$ harmonics can be formulated as follows:

$$\left\{\Delta_k\left(f; f_c\right) | k = 1, \cdots, K, f_c \in \left\{f_c^{(1)}, \cdots, f_c^{(F)}\right\}\right\},  \tag{4}$$

where $F$ is the frequency bin, and thus results in $K \cdot F$ harmonic filters in total.

For the discriminator architecture with the above harmonic tensor, we utilize the combinations of the depthwise separable convolution (DSConv) and pointwise convolution (PConv). Specifically, we treat the number of harmonics $K$ as the channel axis, and first use a DSConv with several kernel filters to learn the harmonic information individually. Then, a PConv is utilized to concatenate each harmonic, which learns the latent representations among harmonics. Finally, we follow the similar encoder network with SoundStream (Zeghidour et al., 2021) to use three EncoderBlocks and one

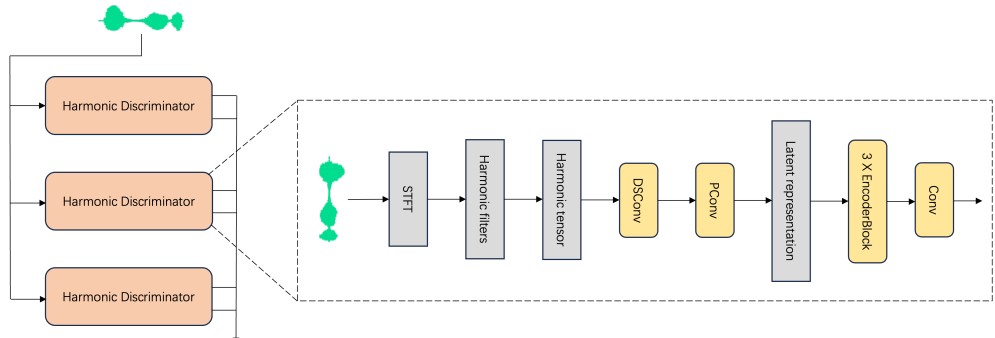

Figure 2: The overall architecture of the proposed multi-scale harmonic discriminator. Each harmonic discriminator means an input of the specific number of harmonic tensor with shape $[K, F, T]$.

convolution layer for feature matching and output, respectively. More details can be found in Appendix A. To further improve the performance, as shown in Figure 2, we apply a multi-scale strategy for the proposed harmonic discriminator, i.e., the number of harmonics is set as three different values, and apply the harmonic discriminator architecture, respectively. Our experiments also validate the effectiveness of this strategy. The multi-scale harmonic discriminator is employed for the GAN-based branch training. In addition, since each harmonic tensor can be designed for the characteristics of a specific frequency range and the bandwidth is learnable, the harmonic discriminator improves the ability of dynamic frequency resolution learning and harmonic tracking.

## 3.5 TRAINING OBJECTIVES

In our paper, the training objectives are composed of adversarial loss, feature matching loss and reconstruction loss. We use the least-squares (LSGAN) loss (Mao et al., 2017) for adversarial loss:

$$\mathcal{L}_G = \mathbb{E}_{s,c} \left[ \sum_{n=1}^{N} D_n \left( G\left(s, c\right) - 1\right)^2 \right], \tag{5}$$

$$\mathcal{L}_D = \mathbb{E}_x \left[ \left(D_n\left(x\right) - 1\right)^2 \right] + \mathbb{E}_{s,c} \left[ D_n \left( G\left(s, c\right)\right)^2 \right], \tag{6}$$

where $x$ is the original waveform, $s$ and $c$ indicate the original mel-spectrogram and condition information, respectively. And $D_n$ is the $n$-th discriminator.

For the feature matching loss, we adopt the $L1$ loss between feature maps of each discriminator in terms of the real and synthetic speech as follows:

$$\mathcal{L}_{FM} = \mathbb{E}_{x,s,c} \left[ \sum_{i=1}^{T} \frac{1}{P_i} \| D_n^i\left(x\right) - D_n^i\left( G\left(s, c\right)\right) \|_1 \right], \tag{7}$$

where $D_n^i$ is the $n$-th discriminator feature map of the $i$-th layer, $P_i$ is unit numbers in each layer.

In our paper, the above losses are employed for the discriminators of MPD, MS-STFT and multi-scale harmonic discriminator. Furthermore, we use all discriminators for the GAN-based branch, and the MS-STFT discriminator for CondNet.

For the reconstruction loss, we apply the multi-resolution STFT loss following Yamamoto et al. (2020) for the reconstructed sub-band waveform and final waveform. In our paper, instead of using a single multi-resolution STFT loss, we design a multi-tier multi-resolution STFT loss between the final waveform and ground-truth. Specifically, we use three tiers multi-resolution STFT losses and each tier denotes a sampling rate of speech. In our work, the multi-resolution STFT losses are computed at three different sampling rates, i.e., 24k Hz, 16k Hz and 8k Hz. Furthermore, for the conditional reconstruction network, a single STFT loss (Yamamoto et al., 2020) for magnitude is also used to guarantee more consistency. The summary of losses is in Appendix B.

## 4 EXPERIMENTS

### 4.1 DATASETS

In our experiments, we utilize a clean 500-hour mixture speech dataset for model training. Specifically, these speech samples are created from our intranet sites by extracting the clean vocal track of television drama. All training speech samples are resampled at 24k Hz. The FFT size and hop size are 1024 and 256, respectively. For each speech sample, we set the number of Mel bins as 160, extract log mel-spectrograms and apply normalization. For testing, we use two datasets. Specifically, a 100-utterance test dataset extracting the clean vocal track of television drama has the same distributions with the training dataset, called in-domain (ID) dataset. To evaluate the generalization ability of model for unseen acoustic conditions, we create another 100-utterance test dataset containing singing voice clips extracted from the Opencpop dataset (Wang et al., 2022), called out-domain (OD) dataset, which has out-of-domain distributions compared with the training dataset.

### 4.2 TRAINING SETUPS

**Generator.** In our proposed DMNet, the number of sub-band is set as 8. For the GAN-based branch, we use three upsample blocks, each contains the hybrid upsample block, post-convolution and MRF module. These upsample blocks achieve 32x upsampling due to predicting 8 sub-bands simultaneously, where up-factor is (4, 4, 2). The post-convolution layer chooses the kernel size 3 and stride 1. The MRF modules have the same configurations with HiFiGAN. The kernel size of transposed convolution is twice of the stride. The output channels of three upsample blocks are 256, 128 and 64, respectively. In addition, the output channels of the first and last convolution layers are 512 and 8, respectively. For CondNet, we adopt 4 ConvNeXt blocks to predict the sub-band Fourier spectral coefficients and the input channel of the first block is converted to 256. The intermediate dimension is 768 and the output channels of the final convolution layers are both 129 for phase and magnitude predicting. Then, we use the inverse STFT with the hop size 32 to generate the low-frequency sub-band waveform. Furthermore, the hop size of STFT is 2 for downsampling. In the residual stack, there are 3 convolution layers and its dilations are 1, 3 and 5. The output channels of three residual stacks are 64, 128 and 256, respectively.

**Discriminator.** We follow the same network of the multi-period discriminator (MPD) in Kong et al. (2020). As for the MS-STFT discriminator, we utilize the similar encoder network architecture with the proposed harmonic discriminator. See Appendix A for the network architecture of MS-STFT discriminator. For the configurations of MS-STFT discriminator, the FFT size, hop size and window size are (1024, 2048, 768), (240, 320, 120) and (960, 1280, 480) in the GAN-based branch, respectively. In the CondNet, the FFT size, hop size and window size are (128, 256, 64), (20, 30, 15) and (128, 256, 64), respectively. For the multi-scale harmonic discriminator, we use 8, 10 and 12 as the numbers of harmonics, respectively. The output channel of the DSConv is the same as the number of harmonics and the output channel of PConv is 32.

**Training.** For the multi-tier multi-resolution STFT loss, three different sampling rates are used to resample the waveform for multi-tier losses computation. Specifically, 24k Hz is the original sampling rate and 3k Hz is applied for the predicted sub-band waveform in the CondNet. More detailed configurations are presented in Appendix C. In addition, we randomly intercept 25600 samples of each speech and apply the batch size as 24 during training process. The weight normalization is employed for all modules. The initial learning rate of generator and discriminator is set as 2e-4 and the model is optimized utilizing the Adam optimizer (Kingma & Ba, 2014) with betas (0.5, 0.9). Furthermore, the learning rate of the proposed generator is halved every 100K steps until 1e-6. For the fast convergence, the generator is pre-trained in the first 20K steps. Finally, we train our proposed model up to two million steps.

### 4.3 BASELINES AND EVALUATIONS

**Baselines.** Three GAN-based methods (HiFiGAN[1] (Kong et al., 2020), TFGAN (Tian et al., 2020) and BigVGAN[2] (Lee et al., 2023)) and two iSTFT-based methods (iSTFTNET (Kaneko et al., 2022)

---

[1]https://github.com/jik876/hifi-gan
[2]https://github.com/NVIDIA/BigVGAN

Table 2: Results of in-domain and out-domain datasets. Models are sorted by the algorithm type, GAN or iSTFT are listed. The best results are listed in bold.

| Method | Type | ID dataset | | | OD dataset | | | Speed (GPU) |
|---|---|---|---|---|---|---|---|---|
| | | PESQ ↑ | MCD ↓ | MOS ↑ | PESQ ↑ | MCD ↓ | MOS ↑ | xRT ↑ |
| Ground Truth | - | 4.500 | 0.000 | $4.52 \pm 0.08$ | 4.500 | 0.000 | $4.33 \pm 0.10$ | - |
| HiFiGAN | GAN | 2.695 | 2.604 | $3.28 \pm 0.18$ | 2.261 | 3.770 | $3.31 \pm 0.19$ | 122.14 |
| TFGAN | GAN | 2.336 | 2.799 | $3.21 \pm 0.16$ | 2.097 | 3.958 | $3.19 \pm 0.17$ | 155.28 |
| BigVGAN | GAN | **3.854** | **1.891** | $\mathbf{3.83 \pm 0.14}$ | **3.565** | 1.984 | $\mathbf{3.74 \pm 0.15}$ | 21.13 |
| iSTFTNET | iSTFT | 2.791 | 2.697 | $3.39 \pm 3.12$ | 2.318 | 3.601 | $3.35 \pm 0.20$ | 170.32 |
| Vocos | iSTFT | 3.622 | 2.087 | $3.73 \pm 0.16$ | 3.349 | 2.100 | $3.64 \pm 0.18$ | **594.05** |
| **DMNet** | - | 3.693 | 1.915 | $3.77 \pm 0.14$ | 3.432 | **1.937** | $3.70 \pm 0.16$ | 102.46 |

and Vocos[3] (Siuzdak, 2024)) are used as baselines. We retrain all baselines using public official codes other than TFGAN and iSTFTNET. In addition, we also utilize unofficial implementations for TFGAN[4] and iSTFTNET[5] training.

**Evaluations.** In our experiments, we utilize both objective and subjective evaluations for our proposed model and baselines. For the objective evaluations, we adopt the Perceptual Evaluation of Speech Quality (PESQ) (Rix et al., 2001) and the mel-cepstral distortion (MCD) (Kubichek, 1993) with dynamic time warping to estimate all models between the synthesized waveform and the ground-truth. For the subjective evaluation, we rely on the 5-point Mean Opinion Score (MOS) metric to estimate the speech quality and intelligibility of test datasets. To assess inference speed, an NVIDIA V100 GPU is used to generate a batch of 24 speech samples (1-second for each sample) and the xRT value that means the speed factor relative to real-time is used for speed evaluation. Value 1.0 of xRT denotes real-time speed.

## 4.4 RESULTS

### 4.4.1 MODEL PERFORMANCE

We first evaluate the performance of our proposed DMNet model compared to the GAN-based and iSTFT-based models, as illustrated in Table 2. In terms of most of the evaluations, our proposed DMNet realizes the superior performance compared to the other baseline models that own the same level parameters. In addition, although the state-of-the-art high capacity GAN-based model BigVGAN achieves the highest evaluation scores in the in-domain dataset, DMNet still remains the comparable performance and bridges the gap in the out-domain dataset, e.g., it achieves the highest MCD score. Moreover, the proposed DMNet accomplishes this task with only 14.2 M parameters in the generator. In contrast, BigVGAN owns 112 M parameters, making it approximately 8 heavier in size compared to DMNet. This characteristic makes DMNet suitable in end-to-end text-to-speech and voice conversion training scenarios. Furthermore, in the out-domain dataset, DMNet notably achieves more competitive performance compared to all baseline models, which verifies the superior generalization capability of the proposed model.

The spectrogram visualization of a singing voice sample in the out-domain dataset is presented in Figure 3. Specifically, we compare the reproduced spectrograms of the typical GAN-based HiFiGAN and the advanced iSTFT-based Vocos. Aliasing artifacts are obviously observed in GAN-based methods. Furthermore, Vocos, correctly handling phase wrapping, makes an improvement over HiFiGAN. Our DMNet achieves a more superior prediction of harmonic structures and contributes to better fast-changing harmonic tracking, due to the using of multi-band processing and multi-scale harmonic discriminator.

For the inference speed, we compare our proposed DMNet to all baseline models. As shown in Table 2, DMNet achieves fast and comparable inference speed than all GAN-based models. Specifically, DMNet realizes slightly slower than TFGAN and HiFiGAN, and approximately 5 times faster than

---

[3]https://github.com/gemelo-ai/vocos

[4]https://github.com/rishikksh20/TFGAN

[5]https://github.com/rishikksh20/iSTFTNet-pytorch

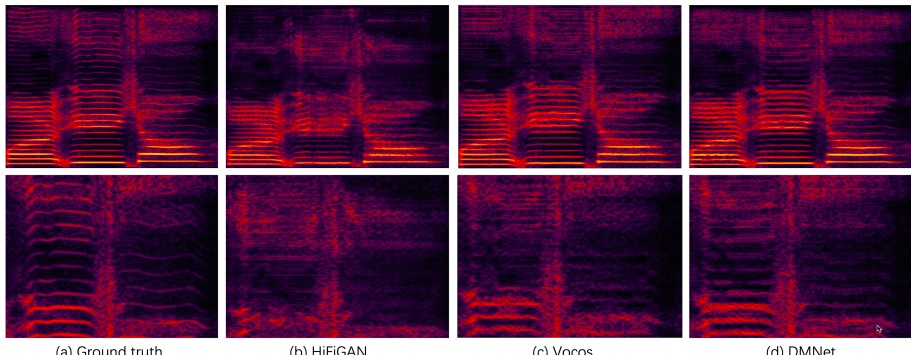

|          |          |          |          |
| :------: | :------: | :------: | :------: |
| (a) Ground truth | (b) HiFiGAN | (c) Vocos | (d) DMNet |

Figure 3: The spectrogram visualization of an out-of-domain singing voice sample. The zoomed-in medium-high frequency view is correspondingly presented at the bottom row.

Table 3: The ablation study of the generator in terms of the objective evaluations (PESQ and MCD).

| Method | ID dataset | | OD dataset | |
| :---: | :---: | :---: | :---: | :---: |
| | PESQ ↑ | MCD ↓ | PESQ ↑ | MCD ↓ |
| **DMNet** | **3.693** | **1.915** | **3.432** | **1.937** |
| w/o post-convolution | 3.689 | 1.917 | 3.422 | 1.941 |
| w/o MRF module | 3.602 | 1.997 | 3.214 | 2.011 |
| w/o hybrid sample | 3.691 | 1.919 | 3.403 | 1.962 |
| w/o sub-band Fourier | 3.599 | 2.023 | 3.221 | 2.099 |
| w/o CondNet | 3.421 | 2.211 | 3.068 | 2.276 |

BigVGAN. In addition, DMNet also owns the comparable inference speed compared to iSTFTNET. This is mainly because of the multi-band processing in the dual-branch network instead of the full-band. While DMNet is approximately 6 times slower in the inference speed compared to Vocos, it achieves more superior performance than Vocos for both objective and subjective evaluations. Therefore, DMNet achieves a better balance between performance and inference speed.

### 4.4.2 Ablation Study

To verify the effectiveness of the proposed components, we conduct the ablation study experiments based on components ablated in this section.

**Generator Ablation.** Results of the generator ablation are shown in Table 3. For the GAN-based branch, we drop the post-convolution layers and the results slightly decline. We also try to add the sinusoidal activation like TFGAN that is not illustrated in Table 3, but it does not result in performance gains. Substituting the MRF modules with residual stacks in TFGAN yields a dramatic performance decline, which verifies the advantages of MRF module. For the sampling modules, we replace the hybrid upsample and downsample modules with normal upsample and downsample convolution layers, i.e., without repeat-convolution and skip-convolution modules, which also leads to slightly degraded quality. This indicates that the hybrid sample modules can effectively alleviate the sampling distortion. In addition, for the CondNet, instead of predicting the sub-band Fourier spectral coefficients, we predict the full-band coefficients and apply the PQMF to obtain the sub-band waveform. This formulation forces CondNet to learn complex multi-periodicity components in the full-band, resulting in a giant performance decline and further verifying the effectiveness of sub-band operator. Furthermore, we directly omit the CondNet, which causes no access to the inductive time-frequency bias, resulting in dramatically degraded quality. This finding also demonstrates the importance of time-frequency structure.

**Discriminator Ablation.** As the number of harmonics $K$ increasing in the discriminator, the results are illustrated in Figure 4. V0 stands for the results without harmonic discriminator. V1-V3 means that only one harmonic discriminator is used and the numbers of the harmonics are 8, 10 and 12, respectively. V4 is the result of the multi-scale version combining the above three harmonic

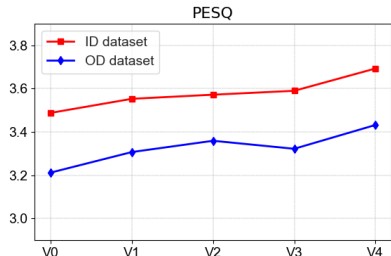 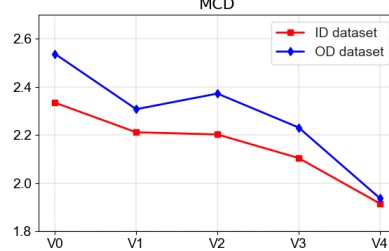

Figure 4: The ablation study of multi-scale harmonic discriminator in terms of harmonic numbers.

Table 4: The ablation study of the using loss in terms of the objective evaluations (PESQ and MCD).

| Method | ID dataset | | OD dataset | |
|---|---|---|---|---|
| | PESQ ↑ | MCD ↓ | PESQ ↑ | MCD ↓ |
| **DMNet** | **3.693** | **1.915** | **3.432** | **1.937** |
| w/o multi-tier loss | 3.681 | 2.002 | 3.299 | 2.032 |
| w/o magnitude loss | 3.688 | 1.934 | 3.403 | 1.966 |
| w/ time domain loss | 3.691 | 1.915 | 3.419 | 1.941 |

discriminators. From Figure 4, we can find that any individual harmonic discriminator is the suboptimal choice for objective evaluations. Furthermore, omitting the harmonic discriminator leads to the dramatic performance declines, especially for MCD metrics both in in-domain and out-domain datasets. Lacking of the ability of fast-changing harmonic tracking is the primary reason. This finding highlights the importance of dynamic frequency resolution in the discriminator. In addition, we also try a smaller or larger $K$, which results in a worse performance. The smaller $K$ reduces the ability of harmonic tracking. Instead, the larger $K$ leads to the loss of fundamental frequency. For example, if $K$ is set as 15, the fundamental frequency higher than 800 Hz is unable to be learned, which results in the degraded performance.

**Loss Ablation.** We also conduct some ablation studies in terms of losses ablated and replaced. The objective results are shown in Table 4. Specifically, we substitute the proposed multi-tier multi-resolution STFT loss with the original version, i.e., only the original sampling rate 24k Hz is applied. The results are shown in the second row in Table 4. This manipulation results in performance declines to a certain extent, especially for PESQ metric, which verifies that the multi-tier multi-resolution STFT loss can facilitate fine-grained reconstruction of waveform and further improve the speech quality. Furthermore, we directly omit the magnitude loss in CondNet, which results in slight performance declines for both PESQ and MCD metrics. This finding indicates that the intermediate consistency constraint of reconstructed sub-band magnitude contributes to the better speech quality. In addition, in our experiments, we also add a time domain loss named multi-scale dynamic loss that is proposed in TFGAN (Tian et al., 2020). However, this time domain loss did not lead to performance gains and decelerates the training speed. A better balance between the time domain loss and other losses may need to be designed, which will be explored in the future work.

## 5 CONCLUSION

In this paper, we propose a dual-branch multi-band neural vocoder with multi-scale harmonic discriminator. Specifically in our model, GAN-based branch and iSTFT-based branch are integrated into a framework. In addition, we utilize the multi-band processing in the dual-branch. We propose the CondNet to predict the sub-band Fourier spectral coefficients, which simplifies the difficulty of model training and further improves the inference speed. Moreover, to more correctly learn the harmonic structures in speech, we design a novel harmonic discriminator, which uses the dynamic frequency resolution to train network and achieves the dramatic performance gains. Our experiment results also demonstrate the effectiveness of the proposed model. In conclusion, DMNet provides a new idea for the advancement of neural vocoders.

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

Figure 5: The overall network architecture of MS-STFT and harmonic discriminator. Substituting the first Conv2D with DSConv and PConv and adding the harmonic filters behind STFT operator form the harmonic discriminator. N denotes the output channel of each module and S is the stride of convolution layer. K denotes the number of harmonics.

## A  HARMONIC AND MS-STFT DISCRIMINATOR NETWORK

We utilize the similar network architecture with SoundStream (Zeghidour et al., 2021), as shown in Figure 5. Specifically for the harmonic discriminator, we first apply the STFT operator for the input waveform, and then use the harmonic filters to get harmonic tensors which are sequentially fed into DSConv and PConv. Furthermore, we use three encoder blocks and each contains two convolution units. For each encoder block, the output channel is exponentially increased and the stride is (4, 2) for frequency and time, respectively. Remarkably, the stride of harmonic discriminator is set as (2, 2) for CondNet. In addition, each output of ConvUnit is used to compute the feature matching loss. As for MS-STFT discriminator, we omit the harmonic filters and replace DSConv and PConv with a 2D-Conv to get the latent representations.

## B  TOTAL MODEL LOSS

A single STFT loss contains the spectral convergence loss and log STFT magnitude loss, which can be written as follows:

$$\mathcal{L}_{sc} = \frac{\| \, |STFT(x)| - |STFT(\widetilde{x})| \, \|_F}{\| \, |STFT(x)| \, \|_F}, \tag{8}$$

$$\mathcal{L}_{mag} = \frac{1}{D} \| log \, |STFT(x)| - log \, |STFT(\widetilde{x})| \, \|_1, \tag{9}$$

where $\widetilde{x}$ is the predicted waveform and $x$ is the ground-truth waveform. $D$ is the number of elements in $|STFT(\cdot)|$.

Therefore, the multi-resolution STFT objective function with $N$ single STFT losses that have different FFT size, hop size and window size can be defined as follows:

$$\mathcal{L}_{MR} = \mathbb{E}_{x,\widetilde{x}} \left[ \frac{1}{N} \sum_{n=1}^{N} \left( \mathcal{L}_{sc}^n \left( x, \widetilde{x} \right) + \mathcal{L}_{mag}^n \left( x, \widetilde{x} \right) \right) \right]. \tag{10}$$

Furthermore, our multi-tier multi-resolution STFT loss can be defined as follows:

$$\mathcal{L}_{MTMR} = \mathcal{L}_{MR24k} + \mathcal{L}_{MR16k} + \mathcal{L}_{MR8k} + \mathcal{L}_{MR3k}, \tag{11}$$

where 24k, 16k, 8k and 3k denote the sampling rates. Notably, 3k is used for the predicted sub-band waveform in CondNet. We merge it into the multi-tier multi-resolution STFT loss for convenience.

In addition, the magnitude loss $\mathcal{L}_M$ in CondNet is defined as follows:

$$\mathcal{L}_M = \mathcal{L}_{sc} + \mathcal{L}_{mag}. \tag{12}$$

The final adversarial loss, feature matching loss and reconstruction loss can be written as follows:

$$\mathcal{L}_{ADV} = \lambda_1 \left( \mathcal{L}_{G-M} + \mathcal{L}_{G-S} + \mathcal{L}_{G-H} + \mathcal{L}_{G-SC} \right) + \\ \mathcal{L}_{D-M} + \mathcal{L}_{D-S} + \mathcal{L}_{D-H} + \mathcal{L}_{D-SC}, \tag{13}$$

$$\mathcal{L}_{FM} = \lambda_2 \mathcal{L}_{FM-M} + \lambda_3 \left( \mathcal{L}_{FM-S} + \mathcal{L}_{FM-H} + \mathcal{L}_{FM-SC} \right), \tag{14}$$

$$\mathcal{L}_{REC} = \lambda_4 \mathcal{L}_{MTMR} + \lambda_5 \mathcal{L}_M, \tag{15}$$

where "$*-M$" denotes the MPD and "$*-H$" stands for the harmonic discriminator. "$*-S$" and "$*-SC$" denote the MS-STFT discriminator using in GAN-based branch and CondNet, respectively. $\lambda_1$, $\lambda_2$, $\lambda_3$, $\lambda_4$ and $\lambda_5$ are 10, 10, 40, 5 and 10, respectively.

Therefore, the final loss $\mathcal{L}$ can be written as follows:

$$\mathcal{L} = \mathcal{L}_{ADV} + \mathcal{L}_{FM} + \mathcal{L}_{REC}. \tag{16}$$

## C  MULTI-TIER MULTI-RESOLUTION STFT LOSS

The detailed configurations of different sampling rates and resolutions are shown in Table 5. The sampling rate 3k Hz is applied for the sub-band waveform in CondNet.

Table 5: The configurations of hierarchical multi-resolution STFT loss for different sampling rates.

| Resolution | 24K | | | 16K | | | 8K | | | 3K | | |
|---|---|---|---|---|---|---|---|---|---|---|---|---|
| | FFT | Hop | Window | FFT | Hop | Window | FFT | Hop | Window | FFT | Hop | Window |
| Resolution1 | 2048 | 240 | 960 | 1024 | 160 | 640 | 768 | 120 | 480 | 256 | 30 | 256 |
| Resolution2 | 1024 | 160 | 640 | 768 | 120 | 480 | 512 | 80 | 320 | 128 | 20 | 128 |
| Resolution3 | 512 | 120 | 480 | 512 | 80 | 320 | 384 | 40 | 160 | 64 | 15 | 64 |

