# OpenReview forum: "A Dual-branch Multi-Band Neural Vocoder with Harmonic Discriminator for High-Fidelity Speech Synthesis"
_ICLR.cc/2025/Conference — ICLR 2025 Conference Withdrawn Submission_

### Official Review · Reviewer_HDCt · 2024-10-20

**Soundness:** 1
**Presentation:** 2
**Contribution:** 1
**Rating:** 3
**Confidence:** 4

**Summary:**

The paper presents a Dual-branch Multi-Band Neural Vocoder with Harmonic Discriminator to improve the harmonic reconstruction of audio waveforms. However, based on the provided results, it does not perform as well as claimed.

**Strengths:**

The author proposes a new harmonic discriminator, which may be beneficial for harmonic reconstruction in audio waveforms.

**Weaknesses:**

line079-119 The last paragraph of the introduction is difficult to understand. The innovations and contributions should be less overlapping and presented in a more itemized and organized manner for better clarity.

line 153 "Unfortunately, the multi-periodicity components are always existent in the full-band signals, bringing challenges in the harmonic modeling and aliasing artifacts elimination." I believe that "aliasing artifacts" is a drawback of the neural network structure. It is not caused by multi-periodicity components.

In Figure 1, line 171, the STFT (Downsample x2) block appears to be incorrect.

In Figure 3, all reconstructed spectrograms generate horizontal lines in the upper half part. Thus, DMNet may not solve the harmonic problem as stated. Moreover, both the objective and subjective evaluations are not satisfactory enough.

The description of the harmonic discriminator is lacking in some details. For instance, there is a need for more elaboration on how the harmonic bands are extracted according to the f_c and f_b. Additionally, it is unclear how the channel mismatch is addressed since each subband is of a different channel.

**Questions:**

Address the concerns raised in the aforementioned weakness.

---

### Official Review · Reviewer_3F9a · 2024-10-29

**Soundness:** 2
**Presentation:** 2
**Contribution:** 1
**Rating:** 3
**Confidence:** 4

**Summary:**

This paper proposes dual-branch multi-band networks that reduce the complexity of time-domain modeling by incorporating iSTFT-based efficient waveform modeling and multi-band conditioning. Additionally, the authors introduce a harmonic discriminator that leverages learnable harmonic filters across multiple scales to enhance the modeling capability of harmonic structures.

**Strengths:**

The paper integrates iSTFT-based waveform modeling from Vocos with HiFi-GAN, efficiently conditioning multi-band features.

**Weaknesses:**

[W1. ConvNeXt]

EVA-GAN has already adopted ConvNeXt-based feature extraction before generating high-resolution of waveform-level features. It would be better if you could add the ablation study using ConvNeXt-based low-level feature extraction, not conditioning iSTFT-based multi-band features to demonstrate the effectiveness of this structure.

[W2. Hybrid Upsample Block]

FA-GAN introduced a similar upsampling block designed to reduce checkerboard artifacts during upsampling.

Shen, Rubing, Yanzhen Ren, and Zongkun Sun. "FA-GAN: Artifacts-free and Phase-aware High-fidelity GAN-based Vocoder." Interspeech, 2024.

[W3. Performance]

Although the authors stated that their model is more efficient, BigVGAN demonstrates better overall performance. Additionally, BigVGAN-v2 has been released with CUDA kernel support, enabling faster inference speeds.

[W4. Training Time]

The paper utilizes multiple discriminators, which results in slow training times. Adding the training times for each model in Table 2 would improve clarity.

[W5. Harmonic filters]

[1] proposed CQT-based harmonic filters for discriminators. Please describe the difference between them and add the ablation study for CQT-based Discriminators.

[1] Y. Gu, X. Zhang, L. Xue and Z. Wu, "Multi-Scale Sub-Band Constant-Q Transform Discriminator for High-Fidelity Vocoder," ICASSP, 2024

**Questions:**

Minor Comments: GAN/iSTFT
It appears that the paper overgeneralizes baseline models in Table 1. Specifically, “GAN” should be replaced with “waveform-level modeling or other word,” as most iSTFT-based models also utilize GANs. Additionally, various vocoders employ sub-band harmonic structures in generators, such as AVOCODO, and dynamic resolution-based discriminators, like multi sub-band discriminators.

[Audio Samples]
I can not find any audio samples. Where is the demo link?

---

### Official Review · Reviewer_dz6f · 2024-11-03

**Soundness:** 3
**Presentation:** 2
**Contribution:** 3
**Rating:** 5
**Confidence:** 4

**Summary:**

The present work proposes DMNet, a GAN-based neural vocoder for speech synthesis making use of multi-band processing. The authors propose two main novel components: a complex-valued spectrogram prediction network used for conditioning ("CondNet"), and a harmonic discriminator component making use of a learnable bandpass filterbank with dynamic frequency resolution. The authors find competitive quality and inference speed compared to other recent neural vocoders such as BigVGAN, iSTFTNET and Vocos, and conduct informative ablation experiments on their proposed components.

**Strengths:**

The method is reasonably original, building upon existing literature and extending it, and the paper shows the importance of the proposed components with solid ablations. The paper is largely well-written and understandable. The harmonic discriminator idea is interesting and seems to work well in improving metric values, as proven in an ablation study (Fig. 4). The authors put in a commendable effort in retraining all compared baselines with their own dataset setup, something that is often lacking in other studies.

**Weaknesses:**

* While the work is solid and well-motivated, the novelty is somewhat limited by being an extension and modification of existing works. I do not strongly hold this against the paper overall however, as the method also exhibits a decent novel position on the quality-runtime tradeoff.

* The experimental evaluation shows competitive results and the model strikes a decent balance between runtime and quality somewhere between BigVGAN and Vocos, but when comparing specifically against Vocos (Siuzdak, 2024) in Table 2, the quality improvements over Vocos seem relatively small for an inference runtime that is ~6x as long as Vocos.

* The listening experiment behind the MOS evaluation (line 399-400) is not well described, only using a single sentence to state that the authors use a 5-point MOS scale. In my opinion, it is necessary to add relevant details such as the platform/service used for evaluation, the number of listeners, the number and kind of samples presented to each listener, and details on the setup such as whether the listening experiment was comparative (multiple methods shown at once) or only a single example was presented at a time.

* While the authors make a convincing case for using a multi-scale harmonic discriminator through their ablation study (Figure 4), it is unclear from the conducted experiments whether the learnability (lines 244-245) of the proposed filterbank was actually necessary to achieve these results. Since the other possible key ingredients are the multiscale nature and the dynamic frequency resolution of the filterbank, the authors should compare their learnable filterbank with a fixed filterbank that also has a dynamic frequency resolution and is used in a multiscale manner, for instance as a constant-Q transform.

* Regarding the dataset (section 4.1), this work solely uses an unusual internal dataset extracted from television drama but does not provide a good reason for why this data was used, rather than more typical and publicly available speech datasets. This choice limits the plausibility of the qualitative placement compared to the baselines. It would be better to also conduct - at least - a small study on more usual datasets that published baseline works make use of.

* Some paragraphs use unusual or unclear grammatical constructions or words, e.g., "The last but the most important" (line 51), "well-existed" (line 64), "resulting in the principal values wrapping across all phase angles" (line 73), "we consider the harmonic as channel axis" (line 107), "[...] as the center frequency $f_c$ goes higher, the bandwidth $f_w$ goes wider, which means a higher time resolution [...] and further improves the tracking ability" (lines 245f.), "the whole dual-branch generator network is trained in the form of sub-band" (line 228), or "other baseline models that own the same level parameters" (line 410). I would recommend using a modern grammar and spell correction software to spot and correct these issues, as the paper's clarity currently suffers from this writing.

* The extent of the observed performance degradations as "giant" or "dramatic" in the ablation study (section 4.4.2) seems overstated at differences of 0.1-0.2 PESQ; I would suggest softening the wording somewhat here.

**Questions:**

* Is my understanding correct that CondNet only predicts the lowest sub-band, i.e., between 0 and 1*12kHz/8 = 1.5kHz, so only rather low frequencies? If so, would the authors agree that this construction is very speech-specific and could be suboptimal (e.g. in music important frequency content occurs far beyond 1.5kHz, and even for speech, not all information relevant for understanding is present below 1.5kHz)? I can see that the authors compare against CondNet predicting *all* sub-bands in section 4.4.2 and find performance degradation for all-band prediction, but have the authors also experimented with predicting more than 1 but fewer than all sub-bands, instead?

* In Section 3.5, the purpose of the additional multi-tier nature of the multi-scale STFT loss is unclear to me. For instance, downsampling the signal from 24kHz to 12kHz and then applying the STFT with halved FFT and hop size (as shown in Table 5 in the appendix) will lead to the same STFT spectrum as before, just with the top half missing, and hence the errors in these lower bands used for the loss are also the same. The same thing then seems achievable by simply weighting lower bands more highly in an STFT loss without a multi-tier nature - am I missing something?

* In line 70, what do the authors mean by "apply[ing] the iSTFT [...] is a challenging area of waveform reconstruction"? Generally, the iSTFT is a well-established operation and not challenging, right?

* In line 212, what is meant by "perform the STFT operator for 2x downsampling"? Generally, non-learned downsampling is done by decimation (and possibly FIR filtering) as far as I'm aware - how does one instead use the STFT for downsampling?

* What do the authors mean by their harmonic discriminator being a "first-of-its-kind approach" (line 236)? Learnable bandpass filterbanks were published before for various purposes, such as the works the authors also cite in the following paragraph (lines 237-238). If the authors simply mean the use of such a filterbank as a discriminator, I find the wording "first-of-its-kind" to be somewhat overstating the novelty.

* In lines 286f., the authors describe the multiscale idea for their discriminator but do not make relevant citations such as EnCodec (Défossez et al, 2022) that employ the same kind of multiscale idea for a discriminator. Please add these relevant citations here.

* In section 4.2, I was missing details on how $\alpha, \beta, \sigma$ are initialized for training the discriminator filterbank - please add these details for reproducibility.

* Also in section 4.2, the authors state that "the weight normalization is employed for all modules", but it is unclear which weight normalization, and for what modules; please add this detail, in the appendix if necessary for space.

* In lines 400-403, the authors describe that they evaluate the real-time factor with a batch of 24 1-second samples. Would the authors say that this is a typical/relevant type of practical scenario? I could imagine that a batch of a single, longer ($\geq$ 10-second) sample would be more relevant for many users of vocoders. I would suggest calculating and showing a similar RT factor on single longer utterances in the Appendix and referring to it in the main text.

* In line 410, the authors state that their method performs better than "other baseline models that own the same level parameters". Could you show me where I can find the actual number of parameters of each compared model, and if it is not yet listed in the paper, add these numbers to support this claim?

Typos / grammar / wording:

* line 459: "owns the comparable inference speed" -> "has a comparable inference speed"
* Eq. (5): I think the notation should be $(D_n(G(s,c)) - 1)^2$ rather than $D_n(G(s,c) - 1)^2$?
* What is meant by "Other than" in line 90/91? Do the authors mean "Instead of", or "besides", in this context?
* Line 413: "highest MCD" -> "best MCD" (it's actually lowest for MCD in Table 2, which is best for the MCD metric, but "lowest" could be confusing too)

---

### Official Review · Reviewer_Ks9N · 2024-11-04

**Soundness:** 1
**Presentation:** 2
**Contribution:** 2
**Rating:** 3
**Confidence:** 4

**Summary:**

This paper proposes DMNet, a neural vocoder that uses both time and frequency domains to generate audio signals. It synthesizes multi-band audio signals with a PQMF filter bank and has an additional module for estimating STFT coefficients for the first sub-band. These coefficients are converted back to the time domain with inverse STFT and then passed to the main branch of the vocoder. The paper also proposes a harmonic discriminator to improve the harmonic structure in generated waveforms.

**Strengths:**

The idea of using both time and frequency domains for neural vocoding is interesting.

**Weaknesses:**

Some technical claims are inaccurate, e.g. lines 234-236: “Therefore, we design a novel harmonic discriminator with dynamic frequency resolution by applying the learnable band-pass filters, which is a first-of-its-kind approach.” This is incorrect, the proposed harmonic discriminator does not have dynamic frequency resolution - the frequency resolution is fixed and determined by the STFT parameters. Generally section 3.4 is confusing, as it reintroduces an approach from [1] but omits key, intuitive details. Specifically, the harmonic filter merely applies a sequence of weighting functions to the magnitude spectrum (similar to the Mel scale). The term “learnable band-pass filter” is misleading and could imply signal processing filtering with phase response, which is not the case here.

The proposed harmonic discriminator is instantiated multiple times with different numbers of harmonics K for a supposedly multi-scale approach. However, if $f_{min}$ is always set to 32.7 Hz, it essentially creates the same sequence of frequency bins, just extending higher for larger K (if not limited by the Nyquist criterion). Applying many of these has nothing to do with multi-scale, and any reported improvements likely come from the increased parameter count of the discriminator.

The idea of combining time and frequency domains is interesting, but not entirely new in audio synthesis. It was introduced in TFNet [2], which would be a good reference to include. TFNet also had some good ideas that could help improve the dual-branch vocoder approach in this paper.


The effectiveness of the proposed method is questionable. The evaluation is limited, and subjective scores for BigVGAN, Vocos, and DMNet largely overlap. No audio samples are provided, and from Figure 3, it appears that DMNet also suffers from periodicity artifacts at higher frequencies. The claim that “DMNet achieves a more superior prediction of harmonic structure” might be an overstatement.

[1] Won, Minz, et al. "Data-driven harmonic filters for audio representation learning."

[2] Lim, Teck Yian, et al. "Time-frequency networks for audio super-resolution."

**Questions:**

1. Fourier-based vocoders are known for being fast, but time-domain vocoders still offer better quality. What is the motivation of adding spectral features to a time-domain vocoder? Shouldn't it be the other way around - improving the quality of Fourier-based vocoders with additional (and possibly lightweight) features with the time-domain signal?

2. The analysis of CondNet's STFT is missing. After training, it should be possible to inspect the generated magnitude and phase spectrograms from CondNet. Did you explore this to see if it produces useful outputs that contribute to the synthesis?

3. Could you clarify what you mean by dynamic resolution in your discriminator, particularly in lines 245-247? You state “as the center frequency f_c goes higher, the bandwidth f_w goes wider, which means a higher time resolution in the high-frequency band and further improves the tracking ability of fast-changing harmonics.” How can time resolution change in this case, since it is determined by the magnitude spectrogram's window and hop size?

4. What is the rationale behind using different numbers of harmonics (K) if f_min is always set to 32.7 Hz? Can you explain how this approach is effectively multi-scale?

---

### Note · Authors · 2025-01-07

I have read and agree with the venue's withdrawal policy on behalf of myself and my co-authors.